# Centromedian–Parafascicular and Somatosensory Thalamic Deep Brain Stimulation for Treatment of Chronic Neuropathic Pain: A Contemporary Series of 40 Patients

**DOI:** 10.3390/biomedicines9070731

**Published:** 2021-06-25

**Authors:** Mahmoud Abdallat, Assel Saryyeva, Christian Blahak, Marc E. Wolf, Ralf Weigel, Thomas J. Loher, Joachim Runge, Hans E. Heissler, Thomas M. Kinfe, Joachim K. Krauss

**Affiliations:** 1Department of Neurosurgery, Hannover Medical School, 30625 Hannover, Germany; abdallat@icloud.com (M.A.); ralf.weigel@kabelmail.de (R.W.); Runge.Joachim@mh-hannover.de (J.R.); Heissler.Hans.E@mh-hannover.de (H.E.H.); Krauss.Joachim@mh-hannover.de (J.K.K.); 2Department of Neurosurgery, University of Jordan, Amman 11183, Jordan; 3Department of Neurology, University Hospital Mannheim, 68167 Mannheim, Germany; christian.blahak@ortenau-klinikum.de (C.B.); ma.wolf@klinikum-stuttgart.de (M.E.W.); 4Department of Neurology, Ortenau-Klinikum Lahr-Ettenheim, 77933 Lahr Ettenheim, Germany; 5Department of Neurology, Katharinenhospital, 70174 Stuttgart, Germany; 6Department of Neurosurgery, St. Katharinen Krankenhaus, 60389 Frankfurt, Germany; 7Neurocenter Berne, 3013 Berne, Switzerland; loher@neurobern.ch; 8Department of Neurosurgery, Division of Functional Neurosurgery and Stereotaxy, Friedrich-Alexander University, 91054 Erlangen-Nürnberg, Germany; thomasmehari.kinfe@uk-erlangen.de; 9Center for Systems Neuroscience, 30559 Hannover, Germany

**Keywords:** centromedian–parafascicular complex, deep brain stimulation, functional neurosurgery, pain, ventroposterolateral thalamus

## Abstract

*Introduction:* The treatment of neuropathic and central pain still remains a major challenge. Thalamic deep brain stimulation (DBS) involving various target structures is a therapeutic option which has received increased re-interest. Beneficial results have been reported in several more recent smaller studies, however, there is a lack of prospective studies on larger series providing long term outcomes. *Methods:* Forty patients with refractory neuropathic and central pain syndromes underwent stereotactic bifocal implantation of DBS electrodes in the centromedian–parafascicular (CM–Pf) and the ventroposterolateral (VPL) or ventroposteromedial (VPM) nucleus contralateral to the side of pain. Electrodes were externalized for test stimulation for several days. Outcome was assessed with five specific VAS pain scores (maximum, minimum, average pain, pain at presentation, allodynia). *Results:* The mean age at surgery was 53.5 years, and the mean duration of pain was 8.2 years. During test stimulation significant reductions of all five pain scores was achieved with either CM–Pf or VPL/VPM stimulation. Pacemakers were implanted in 33/40 patients for chronic stimulation for whom a mean follow-up of 62.8 months (range 3–180 months) was available. Of these, 18 patients had a follow-up beyond four years. Hardware related complications requiring secondary surgeries occurred in 11/33 patients. The VAS maximum pain score was improved by ≥50% in 8/18, and by ≥30% in 11/18 on long term follow-up beyond four years, and the VAS average pain score by ≥50% in 10/18, and by ≥30% in 16/18. On a group level, changes in pain scores remained statistically significant over time, however, there was no difference when comparing the efficacy of CM–Pf versus VPL/VPM stimulation. The best results were achieved in patients with facial pain, poststroke/central pain (except thalamic pain), or brachial plexus injury, while patients with thalamic lesions had the least benefit. *Conclusion:* Thalamic DBS is a useful treatment option in selected patients with severe and medically refractory pain.

## 1. Introduction

The treatment of chronic pain syndromes has been a focus of functional neurosurgery since its inception [1,2]. A variety of ablative approaches and neurostimulation procedures have been introduced over the decades targeting both the peripheral and the central nervous system [3,4,5]. While medical treatment of pain has much improved and many pain syndromes are managed satisfactorily with pharmacotherapy including opioids and analgetic drugs such as pregabalin and gabapentin, the treatment of neuropathic and central pain still remains a major therapeutic challenge [6,7]. With regard to the opioid crisis and the need for alternative therapeutic strategies, deep brain stimulation (DBS) for treatment of pain has received renewed interest [8,9].

The thalamus has been recognized early to be involved in pain processing via its relay function for various sensory and pain conducting pathways and its wide connectivity with cortical regions involved in nociceptive neural transmission, including primary sensory, limbic and cognitive-associative domains [10,11]. A variety of thalamic targets were approached with ablative surgery in the 1950s and the 1960s based on different concepts and theories, including the medial dorsal (MD) nucleus of the thalamus, the periventricular gray (PVG), the pulvinar, the centromedian–parafascicular (CM–Pf) nucleus, and others [12,13,14].

In the 1970s, DBS was first applied for the treatment of nociceptive and neuropathic pain, before it became popular as a therapeutic option for movement disorders [15]. The targets used most commonly were the somatosensory thalamic nuclei, the ventroposterolateral (VPL) and the ventroposteromedial (VPM) nucleus, and the PVG [16,17,18,19,20,21,22,23,24,25,26]. Reported results were variable in different series, and in particular in patients with nociceptive pain the beneficial effect often was described as transient with pain recurrence was observed over a period of several months [27]. The popularity of thalamic DBS for the treatment of chronic pain declined when approval of the Federal and Drug Administration (FDA) could not be accomplished after two large trials were considered failures for a variety of reasons [28,29]. Nevertheless, single centers worldwide continued to perform thalamic DBS in patients with various pain syndromes, and a few case series with variable outcomes were published over the past two decades [30,31,32,33,34,35,36,37,38,39,40,41]. Many of these studies, however, were hampered by limited numbers of patients and relatively short follow-up periods. While some studies had a prospective study design, there were no randomized controlled studies, and it has been concluded that further studies are necessary to support the potential value of thalamic DBS as a treatment option in neuropathic pain syndromes [5,42].

We have reintroduced the CM–Pf as a target for the treatment of chronic pain in the late 1990s in patients with refractory neuropathic pain in order to address the affective component of pain perception and processing [43,44,45]. Since then, we have continued to perform thalamic DBS in selected patients with neuropathic pain syndromes according to the initial trial design, inserting electrodes concomitantly in the CM–Pf and the somatosensory thalamus, VPL or VPM contralateral to the body site affected by chronic pain. Here, we present our experience and findings over a period of 20 years in a series of 40 patients outlining the target choice for chronic stimulation and long-term follow-up.

## 2. Materials and Methods

This is a study with retrospective analysis of long-term follow-up. All patients underwent bifocal implantation of DBS electrodes in the CM–Pf and the VPL or the VPM (facial or cranial pain) contralateral to the side of pain. Surgical procedures were performed or supervised by the senior author (JKK) in the Departments of Neurosurgery in Berne (Switzerland), Mannheim and Hannover (Germany) between December 1997 and December 2014.

After thorough interdisciplinary screening of inclusion and exclusion criteria (see below), bifocal thalamic DBS was offered to patients with chronic neuropathic pain syndromes. All patients underwent preoperative standard assessment including MR imaging, evaluation of the pain syndrome with regard to site of pain, and pain intensity according to a visual analogue scale (VAS) from 0 (no pain) to 10 (maximal imaginable intensity of pain) as defined for specific pain determinants, and administration of the Mini Mental State Examination (MMSE) and the Hamilton Depression Rating Scale (HDRS).

The determinants for the specific VAS scores for the intensity of pain were: maximum, minimum and average intensity of pain during the past week, pain at the time of the presentation, and allodynia upon clinical examination.

### 2.1. Patient Selection

Inclusion criteria. To be considered for surgery, patients suffered from medically-refractory chronic neuropathic pain syndromes. A defined cause or lesion in the central or peripheral nervous system underlying the development and the maintenance of the pain syndrome was required. Trials with various analgetic medications including non-steroidal anti-inflammatory drugs, opioids, antidepressants, and gabapentinoids did not yield sufficient relief or were not tolerated because of side effects. Before 2005, patients needed to have at least one trial of medication with gabapentin, and after 2005 a trial with pregabalin. The minimum duration of severe pain was at least one year with a profound negative impact on well-being.

Exclusion criteria. Patients who had a VAS maximum pain score of less than 7 or a VAS average pain score of less than 4 were excluded. Also, a prevalent nociceptive pain type disqualified patients from participation in the study. Patients with dementia, substance abuse, a history of psychiatric comorbidities, ongoing litigation, fibromyalgia, or a short life expectancy were excluded as well.

Patients with neuropathic pain syndromes seeking operative pain therapy who seemed to be better suited for less invasive surgery were offered spinal cord stimulation, dorsal root ganglionectomy or more recently also dorsal ganglion stimulation, and other options [46,47].

The study was conducted according to the guidelines of the Declaration of Helsinki. All subjects provided informed consent. Institutional review board approval was obtained for LFP recording (Beck at al, 2020), but was not required for this study because of its retrospective nature.

### 2.2. Surgical Procedures and Test Stimulation

The surgical procedure and the temporary externalization of electrodes after confirmation of electrode positioning for test stimulation and for recording of local field potentials investigating the role of the CM–Pf complex in attentional processing and goal-oriented behavior selection has been described in detail elsewhere [48,49,50].

Implantation of DBS electrodes. All procedures were performed as awake surgeries without analgosedation. For the stereotactic procedure a Riechert–Mundinger frame with the Zamorano–Dujovny semi-arc was used. The thalamic targets were determined by CT-stereotactic planning guided by preoperative MR imaging. The average stereotactic coordinates for the CM–Pf target were x = 8 mm lateral to the intercommissural line, y = 8 mm posterior to the midcommissural point, and z = 0 at the level of the intercommissural plane. Target coordinates for the VPL were: x = 14, y = −10, and z = 0; and for the VPM: x = 12, y = −10, and z = 0.

During surgery, patients were in the supine position with slight upper body elevation. The stereotactic frame was secured to the operating table. After a C-shaped incision under local anesthesia, two burr holes were made, oriented in an oblique axis just anterior to the coronal suture. A single microelectrode (Leadpoint system, Medtronic Inc., Minneapolis, USA) was inserted along the CM–Pf trajectory in most instances. It was gradually advanced through a guiding cannula with a microdrive, starting from 10 mm above the target with recordings made until 5 mm below target to detect low threshold calcium spikes (LTS) bursts neurons. The length of the DBS electrode along the trajectory was then adjusted according to the microelectrode findings. Thereafter, the quadripolar DBS electrode (model 3387, Medtronic Inc., Minneapolis, MN, USA) was inserted. Macrostimulation was performed to look for the occurrence of stimulation-induced side effects or a sensation of warmth in the CM–Pf. The VPL and the VPM targets were approached, in general, without microelectrode recording. Macrostimulation was used to determine the site and the threshold of stimulation-induced paresthesias. The DBS electrodes were fixed to the skull with the burr hole ring and cap method. After connection of the DBS electrodes to extension wires these were tunneled to the parietal region and externalized. After skin closure, a stereotactic CT was performed to determine the location of the implanted electrodes and to rule out hemorrhagic complications.

Test Phase. Test stimulation of either target was performed via the externalized extension cables for a period of several days (4–10 days) postoperatively. Each target was stimulated at least for a period of two days. Patients were asked daily to rate the intensity of their pain with different stimulation settings according to the specific VAS pain scores. During the test phase, patients received intravenous antibiotic prophylaxis with a cephalosporine, usually cefazolin. After completion of the test phase, the results were discussed extensively with the patient. Whenever improvement of pain was rated greater than 30% on one of the specific VAS scores, and this improvement was considered useful by the individual patient, implantation of a pacemaker for chronic stimulation was suggested choosing the DBS electrode for chronic stimulation which had yielded the better results during test stimulation. Otherwise, in case of limited stimulation response, explantation of the electrodes was proposed.

Pacemaker Implantation. Surgery for implantation of the pacemaker and the extension cables was performed under general anesthesia. First, the extension cables were cut and removed. Via a subclavicular incision, usually on the left side, a subcutaneous or subfascial pocket was created for the implantable pulse generator (IPG). In general, a single channel IPG was implanted, only more recently dual channel IPGs were reimbursed by health care providers (Itrel 2, Kinetra, Synergy, Activa PC or Activa RC, Medtronic Inc., Minneapolis, MN, USA). Then, both DBS electrodes were tunneled via a retroauricular incision. In the early phase of the study a single extension cable was tunneled from the incision site down to the pocket and connected to the IPG, while the other electrode was isolated locally. In the latter phase, two extension cables were tunneled down to the IPG.

### 2.3. DBS Programming and Follow-Up

Initial DBS programming. The IPG was programmed on the first postoperative day after a monopolar review of impedances. For the initial programming a bipolar configuration was chosen using either the two lowest or the two middle contacts setting the pulse width at 210 µs and the frequency at 130 Hz. The voltage was set below the threshold evoking persistent paresthesias at the VPL or the VPM target, usually between 0.5 and 1.5 V, while it was set between 2.0 and 3.0 V at the CM–Pf target.

Follow-up and Outcome Assessment. Patients were followed with clinical appointments at regular intervals postoperatively for adjustment of stimulation settings and assessment of outcome according to a standard protocol. In the early phase patients were seen at 3-month intervals, and then yearly thereafter. Assessment of outcome included rating of the five specific VAS pain scores, the MMSE and the HDRS. In addition, patients determined their subjective perception of benefit according to a self-rating scale as excellent, marked, moderate, minor or none. All changes in medication were noted.

In order to assess long term efficacy follow-up periods were summarized as follows: FU I: early postoperative, 3–11 months; FU II: 12–23 months, FU III: 24–47 months, and FU IV: 48 months and longer.

Adjustments of stimulation settings were made with regard to clinical response. IPGs were replaced preemptively when possible. At the time of IPG replacement, patients were offered the possibility to switch to the other target in case the effect of stimulation had waned. In single instances, a dual channel IPG was offered.

### 2.4. Statistical Analysis

Statistical analysis. Demographic data of patients were summarized using average values and their standard deviations as were the results of the visual analogue scale (VAS). Categorical data were given as frequencies with proportions. Statistical tests were used according to the measurements’ level, including the Kruskal–Wallis test and chi2 test. Statistical analyses were performed using JMP^®^ 15 software (SAS Institute Inc., Cary, NC, USA, 1989–2020). *p*-values < 0.05 were considered statistically significant and indicated by stars: * indicates *p* < 0.05, ** *p* < 0.01, *** *p* < 0.001 by default.

## 3. Results

Forty patients were included in the study. Demographic and clinical data are shown in Table 1. The mean age of the study population at the time of surgery was 53.5 years (range, 24–87 years). The mean duration of pain was 8.2 ± 10 years (range 1–55 years). There was an even gender distribution (20 women, and 20 men). The persisting pain syndromes were classified as: facial pain (trigeminal neuropathic/atypical facial pain) in six patients (15%), complex regional pain syndrome (CRPS) in eight patients (20%), poststroke/central pain (except thalamus) in seven patients (17.5%), central thalamic pain in four patients (10%), postherpetic pain in four patients (10%), deafferentation pain due to spinal cord lesions in four patients (10%), brachial plexus injury in three patients (7.5%), failed back surgery syndrome with neuropathic pain in two patients (5%), neuropathic pain after peripheral nerve lesion in one patient (2.5%), and phantom limb pain in one patient (2.5%). Other pain procedures including spinal cord stimulation, trigeminal ganglion thermodenervation or intrathecal drug pump implantation had been performed in 11 patients earlier.

The mean specific VAS pain scores of all 40 patients before DBS electrode implantation were: 9.0 ± 1.1 (maximum pain), 4.8 ± 2.6 (minimum pain), 7.1 ± 1.6 (average pain), 7.5 ± 1.7 (pain at time of presentation), and 6.9 ± 3.9 (allodynia). The mean MMSE was 27.8 ± 2.3, and the mean HDRS 11.5 ± 5.6.

DBS electrodes were implanted on the right side in 22 patients, on the left side in 17 patients, and one patient (with complete spinal cord transection) had bilateral surgery. LTS neurons could be recorded in the CM–Pf in 28/30 instances. Upon test stimulation of VPL or VPM, contralateral paresthesias could be evoked at a threshold between 0.5 V and 2.0 V in all patients. There were no intraoperative or early postoperative complications. Postoperative imaging confirmed appropriate location of the DBS electrodes in the target structures. There were no instances of postoperative intracranial hemorrhage.

CM–Pf stimulation during the test phase resulted in a highly significant reduction of all five specific VAS pain scores (*p* < 0.01). Likewise, VPL or VPM test stimulation yielded similar reductions in the five specific VAS pain scores (*p* < 0.01). The maximum pain score, for example, was reduced from 9.0 ± 1.1 to 5.0 ± 3.0 (44%) on average during CM–Pf test stimulation, while it was reduced from 9.0 ± 1.1 to 6.3 ± 3.0 (34%) on average with VPL or VPM stimulation. There were no statistically significant differences when the stimulation effects of the two different targets were compared for all five specific VAS pain scores.

Out of the total group of 40 patients, 33 (82.5%) indicated that they had a clear benefit during test stimulation and were thus scheduled for IPG implantation for chronic DBS. The benefit of test stimulation in the 33 responders was thought to be more pronounced in 20 patients with CM–Pf test stimulation, and in 13 patients with VPL or VPM stimulation.

The etiologies of pain syndromes in the seven patients who did not achieve benefit during test stimulation were: central thalamic pain (three), trigeminal neuropathic/atypical facial pain (one), postherpetic pain (one), deafferentation pain due to spinal cord injury (one), and CRPS (one). The highest number of primary non-responders vs. responders was found in the subset of patients with central thalamic pain (three of four patients). All three patients with central thalamic pain who did not respond to test stimulation had thalamic lesions which, however, were not localized in the target areas.

The mean follow-up of the 33 patients who underwent subsequent IPG implantation was 62.8 months (range, 3–180 months). The number of patients who were available for the different follow-up periods and the reasons for attrition are shown in the flowchart in Figure 1. For FU IV, 18 patients were still available: 10 with CM–Pf stimulation, 7 with VPL or VPM stimulation, and 1 with concomitant bifocal stimulation. Overall, the reasons for attrition from the study included the following: three patients died due to unrelated causes, four patients were lost to follow-up, in five patients the stimulation systems were explanted because of chronic infection (in three of them after IPG replacement), and three other patients had not reached the corresponding follow-up periods. All five patients whose stimulation systems were explanted because of infection optioned not to undergo re-implantation again.

Secondary surgeries apart from IPG replacement were performed in a total of 11/33 patients (33%) because of hardware-related complications (eight instances) or for connection of the second implanted electrode because of decline of efficacy with stimulation of the target which had been chosen initially (three instances). As indicated above, 5/33 patients (15.2%) had their complete systems explanted due to infection between one and five years after the primary surgery despite several attempts in three of them to preserve components of the neurostimulation system. Three other patients had lead revisions because of electrode fracture or discomfort because of bowstringing.

The mean stimulation settings at the last follow-up for the CM–Pf target were: 2.5 ± 0.8 V, 131 ± 4.2 Hz, 210 ± 0 µs; and for the VPL or the VPM target: 2.1 ± 1.0 V, 129 ± 5.5 Hz, 207 ± 45 µs.

The development of the five specific VAS pain scores over time for the whole study population when available for the different follow-up periods is shown in Table 2, including the mean percentage of improvement when compared to the preoperative assessment. While the improvement of VAS maximum pain scores paralleled that of VAS average pain scores, the most profound effect was noted for the VAS allodynia pain score. In general, the mean improvement was highest in those patients who were still under active stimulation during FU IV (48 months or longer).

The VAS maximum pain score in the 33 available patients at FU I was improved by ≥50% in 15/33, and by ≥30% in 22/33; in the 30 patients at FU II it was improved by ≥50% in 13/30, and by ≥30% in 18/30; in the 25 patients at FU III it was improved by ≥50% in 12/25, and by ≥30% in 17/25; and in the 18 patients at FU IV it was improved by ≥50% in 8/18, and by ≥30% in 11/18.

The VAS average pain score in the 33 patients at FU I was improved by ≥50% in 17/33, and by ≥30% in 21/33; in the 30 patients at FU II it was improved by ≥50% in 15/30, and by ≥30% in 20/30; in the 25 patients at FU III it was improved by ≥50% in 13/25, and by ≥30% in 17/25; and in the 18 patients at FU IV it was improved by ≥50% in 10/18, and by ≥30% in 16/18. The number of patients who improved by ≥50% and ≥30% classified with regard to the different pain syndromes in the different follow-up periods is shown in Table 3.

The overall outcome with regard to the five specific VAS pain scores for the whole group (CM–Pf and VPL or VPM) during follow-up is shown in Figure 2. On a group level, all changes remained statistically significant over time. When preoperative scores were compared to the scores at the different follow-up periods for the two different targets improvement of the scores was significant at all time points for either target. However, there was no statistical difference when comparing the efficacy of CM–Pf versus VPL or VPM stimulation.

The degree of improvement differed between patients with different pain etiologies, as outlined above. The overall outcome with regard to the five specific VAS pain scores for the three largest groups of patients during follow-up is shown in Figure 3 (facial pain), Figure 4 (CRPS), and Figure 5 (poststroke/central pain (except thalamus)). The least benefit, overall, was seen in patients with central thalamic pain. While in 3/4 patients test stimulation was ineffective as outlined above, the one patient who had moderate improvement with CM–Pf DBS decided not to undergo electrode implantation once more after the neurostimulation system had been removed six months after surgery because of infection.

There were no differences between the preoperative MMSE score (27.8 ± 2.3), and the scores at the different follow-up periods: FU I 27.7 ± 1.5, FU II 27.9 ± 2.3, FU III 28.5 ± 1.1, and FU IV 28.4 ± 1.1. The HDRS, however, was improved significantly at all follow-up periods as compared to preoperatively: preoperative 11.5 ± 5.6, FU I 7.4 ± 6.6, FU II 8.2 ± 4.9, FU III 7.4 ± 4.4, and FU IV 5.3 ± 2.6 (*p* < 0.01, respectively).

No significant correlation was found between the duration of chronic pain and the improvement of any of the five specific VAS pain scores. While all patients were on different combinations of medications preoperatively, the number and the dosages of medication could be reduced in those who benefited from chronic DBS as shown in Figure 6. The percentage of patients without analgetic medication at the time of the preoperative assessment was 0% (0/40), in contrast to the postoperative evaluations with 15% (5/33) at FU I, 20% (6/30) at FU II, 24% (6/25) at FU III, and 28% (5/18) at FU IV.

The scores of patients’ self-rating of overall benefit according to their subjective perception at the different periods of follow-up is shown in Table 4. Although the overall number of patients decreased from the first follow-up to the last follow up, the proportion of patients with excellent/marked improvement (18/33, 54.5%) vs. moderate/minor/no improvement (15/33, 45.5%) at FU I, did not differ significantly at FU IV (11/18, 61.1%) vs. (7/18, 38.9%), and at other follow-up periods (chi^2^, *p* = 0.78).

## 4. Discussion

Our study shows that thalamic DBS can result in sustained improvement of chronic neuropathic pain due to different etiologies at long term follow-up beyond four years. The best results were achieved in patients with brachial plexus injury, facial pain or poststroke/central pain (except thalamus), while DBS produced more inconsistent and less favorable outcome in patients with CRPS, FBSS, postherpetic pain, peripheral nerve lesions or phantom limb pain, and was generally disappointing in those with spinal cord injury or thalamic lesions. Stimulation of the CM–Pf complex was not superior to stimulation of the somatosensory thalamus neither in the acute setting nor upon chronic stimulation. Overall, results of CM–Pf stimulation, however, appear to be comparable to those reported for PVG stimulation [33,36,51,52]. While surgery was tolerated well, the long-term course was burdened by hardware issues in about a third of patients, which outlines the necessity of hardware and electrode improvement in future studies. Furthermore, there was a relatively high rate of infection, and it remains unclear whether or not temporary externalization of DBS electrodes facilitated hardware infection. As with other studies in this field, the estimation of pain improvement using VAS scales and other instruments introduces subjectivity which might result both in underestimation or overestimation of results concerning both patient selection for chronic implantation and long term outcome. In addition, psychological issues and changes in mood need to be considered [53,54] since depression may affect markedly the quality of life independent from pain. Notably, also at long term follow-up several patients in our study were still on antidepressant medication. This aspect in particular needs further consideration in future studies.

While several reviews on various aspects of DBS for chronic pain identifying gaps and controversies have been published [42,55,56,57,58,59,60,61,62,63], the number of recent original publications is very limited. Regarding the heterogeneity of indications, targets, and methods for the assessment of pain, literature reviews came to partially incongruent conclusions and recommendations. In a meta-analysis published in 2005, Bittar and colleagues determined that DBS would be more effective for nociceptive pain than for deafferentaion (neuropathic) pain at long term follow-up (63% versus 47%) [58]. Notably, however, since then indications for DBS in patients with chronic pain have shifted to include mainly those with different forms of neuropathic pain including deafferentation syndromes, facial pain and central pain [63]. In a recent systematic review including 22 articles published since 2001, Frizon et al. noted that poststroke pain, phantom limb pain and brachial plexus injury had become the most frequent indications for DBS [42]. The authors pointed out that the results for the same indications between different studies, however, were very variable, even when apparently using the same target. For example, while PVG stimulation yielded improvement by 40 to 50% in poststroke patients as reported by the Oxford group [36], results were less favorable for this indication in studies published by other groups [31,33]. Another recent systematic literature review on DBS treatment for pain came to the conclusion that only five studies published since 1990 would qualify to meet their inclusion criteria, which required a prospective study design, the inclusion of multiple pain patterns, and follow-up for more than one year [5]. Remarkably, the only two randomized controlled studies on DBS for pain encompass stimulation of the posterior hypothalamus for cluster headache [64], and of the ventral striatum/anterior limb of the internal capsule for poststroke pain [65]. Finally, while all published reviews indicate that DBS for chronic pain is likely effective, they stress the need for more investigation of this subject.

The overall results for the different indications and the various targets used for DBS in chronic pain have been covered well in the published literature reviews [5,42,59,60,61,62,63]. Thus, in the following we will provide only a brief and general review on more recent studies, and stress several issues which are relevant for the interpretation of our data.

In the study with the largest number of patients published thus far, the group from Oxford summarized the outcome of 85 patients who underwent thalamic DBS for various pain syndromes [36]. Of these, 11 patients did not experience improvement of pain during test stimulation and did not proceed with chronic DBS. Remarkably, 15 patients were lost during follow-up or had to be excluded because of several reasons. Of the remaining 59 patients, 39 had sustained global improvement at follow-up. The efficacy of chronic stimulation did not differ in patients with different stimulation sites: 21 patients had PVG stimulation only, 5 had VPL or VPM stimulation, and 13 had combined stimulation of both targets. Combined stimulation of both VPM and periaqueductal grey (PAG) was used in a recent study reporting on a one-year follow-up in a series of seven patients with facial pain [39]. Notably, in that study the length of trial stimulation was extended up to 63 days in single patients, and full improvement of pain was noted only at six to nine months after chronic stimulation. Another recent study reported on three-year follow-up of chronic VPL DBS in a group of 16 patients with either brachial plexus injury or phantom limb pain [66]. In contrast to previous studies, improvement between the two different groups was comparable. Remarkably, similar to our study, there were few statistically significant differences between one and three years of follow-up, and DBS had retained its efficacy in the majority of patients.

In addition to the various thalamic targets which have been used for the modulation of neuropathic pain, other areas within central pain circuitries have been introduced both regarding the primary and the secondary processing of chronic pain [67,68,69]. While several studies focused on modulation of primary sensory network activity [59,70], others targeted areas associated with limbic pain processing such as the anterior cingulum [68] or with partially unknown connectivity such as the motor cortex [63,71,72]. It has been noted that there is a remarkable overlap in the indications for either motor cortex or thalamic stimulation for neuropathic pain, and there is little data to allow a direct comparison between these two methods [30,52,71,72].

Many of the thalamic targets which had been used for thalamotomy in the past were adopted for DBS after its introduction in clinical practice by early pioneers in the 1970s [17,73,74]. Certainly, the thalamic target used most frequently for DBS is the somatosensory thalamic relay nucleus consisting of the VPM and the VPL connecting the neospinothalamic pathways with the somatosensory cortex. As such, information processing in the somatosensory nucleus is implicated in the localization and grading of the severity of pain. The mechanism of VPL or VPM stimulation has been thought to be related to the inhibition of dysfunctional bursting patterned neuronal activity and faulty reorganization within the somatosensory thalamus due to damage of VPL/VPM afferents [75,76]. Stimulation of the PVG was developed after the observation that periaqueductal stimulation could produce analgesia in an animal model [10]. Initially, PVG stimulation was used preferentially for patients with nociceptive pain [24,77]. It was suggested that PVG DBS involves an opioid-like mechanism, but that concept has been contested for various reasons indicating that the effect of PVG stimulation might also be mediated through the CM–Pf complex, the nucleus raphe magnus and the magnocellular region of the nucleus reticularis gigantocellularis [45,78].

The CM–Pf complex constitutes the major portion of the intralaminar thalamus in primates and it is considered to be a central node in the thalamus–basal ganglia–cortex loop [79,80]. It is involved mainly in attentional orienting, and it may function in different modes within its cortical and subcortical networks [50,81]. The CM–Pf complex along with the centrolateral (CL) thalamic nucleus has also been shown to be implicated in pain processing [82]. Animal studies and clinical studies in humans indicate that the CM–Pf complex is part of a medial pain system, which seems to play a major role primarily in the affective and motivational dimensions of pain [45]. Single-unit recordings from the CM–Pf complex have shown that the activity of the CM–Pf cells is modified by painful stimuli. Under pathological conditions, bursting firing patterns and altered discharge rates were found. There is evidence that the CM–Pf complex might also be involved in the mediation of the beneficial effects of somatosensory thalamic stimulation and PVG stimulation [45]. The CM had been an early target for thalamotomy in patients with central pain yielding pain relief of more than 50% in 9 of 18 patients at short term follow-up [14]. DBS of the CM–Pf was pioneered in the late 1970s by Andy and by Ray and Burton [21,23]. Remarkably, Ray and Burton reported more than 50% pain relief in 21 of 28 patients with various pain syndromes [23]. In the 1990s, Young propagated the CM–Pf and other medial thalamic nuclei as targets for gamma knife thalamotomy in patients with chronic pain [83]. After we re-attracted attention to the CM–Pf as a potential target for functional neurosurgery both with regard to movement disorder and chronic pain [43,44,45], there has been limited interest once more in using this thalamic region in clinical practice. More recently, Sims-Williams and colleagues demonstrated that facial pain associated with anesthesia dolorosa was reduced by an average of 67% in a small series of patients with CM–Pf stimulation [84]. When they compared the effect to that of periaqueductal grey stimulation, they found evidence of differing mechanisms resulting, however, in comparable effects on pain reduction. In the present study we could not confirm our earlier observation in a smaller patient collective that the short-term effects of CM–Pf stimulation were superior to those of VPL or VPM stimulation [45].

Given the inter- but also intraindividual variability, it remains largely unknown which thalamic nucleus would represent the optimal target for chronic stimulation. With regard to the proximity of the PVG and the CM–Pf, it has to be considered that stimulation of one of these targets might also affect the other structure, in particular in patients with wide third ventricles. Another structure that might be stimulated with DBS electrodes in the CM–Pf is the neighboring CL [85].

Currently, there are two new but divergent developments which spike interest in thalamic targets for treatment of chronic pain again. First, the recording of local field potentials from target regions used for DBS in movement disorders has allowed not only a better understanding of the underlying pathophysiology of several disorders, but also of the mechanisms involved in the efficacy of DBS [86,87]. Recently, the functional dynamics of thalamic local field potentials have also been unraveled in patients with chronic pain syndromes recorded from the sensory thalamus [88,89,90]. These findings have raised hope that intrinsic signals encoding the somatosensory, affective and cognitive dimensions of pain could be used for the development of closed-loop DBS techniques in patients with chronic pain [91]. Second, along with the introduction of focused ultrasound there is a renaissance of applying lesions in the thalamus also in pain syndromes [92,93,94].

While the strengths of our study are the relatively high patient numbers as compared to previous studies, the inclusion of patients with various etiologies, and the long follow-up, there are also several limitations. First of all, we did not have the option to compare systematically the effect of CM–Pf versus somatosensory stimulation over the course of our study in individual patients. To achieve this, it would have been necessary to implant dual channel pacemakers in all patients which, however, was not possible because of lack of reimbursement from health insurance carriers for such a policy. Second, we did not include a sham stimulation protocol to control for placebo effects. While this would have been feasible for CM–Pf stimulation, it would have been difficult to achieve with somatosensory thalamic stimulation. On the other hand, the prolonged benefit of responders on long-term chronic stimulation after four years and argues against a strong placebo effect.

## 5. Conclusions

Thalamic DBS is a useful treatment option in selected patients with severe and medically refractory neuropathic pain. Bifocal implantation of DBS electrodes is helpful to select the stimulation target in an individual patient. To further determine the role of thalamic DBS, randomized placebo-controlled studies are highly warranted. In view of the heterogeneity of central and neuropathic pain syndromes, we advocate for the conceptualization of multicenter studies applying the same surgical and stimulation protocol and uniform outcome measures.

## Figures and Tables

**Figure 1 biomedicines-09-00731-f001:**
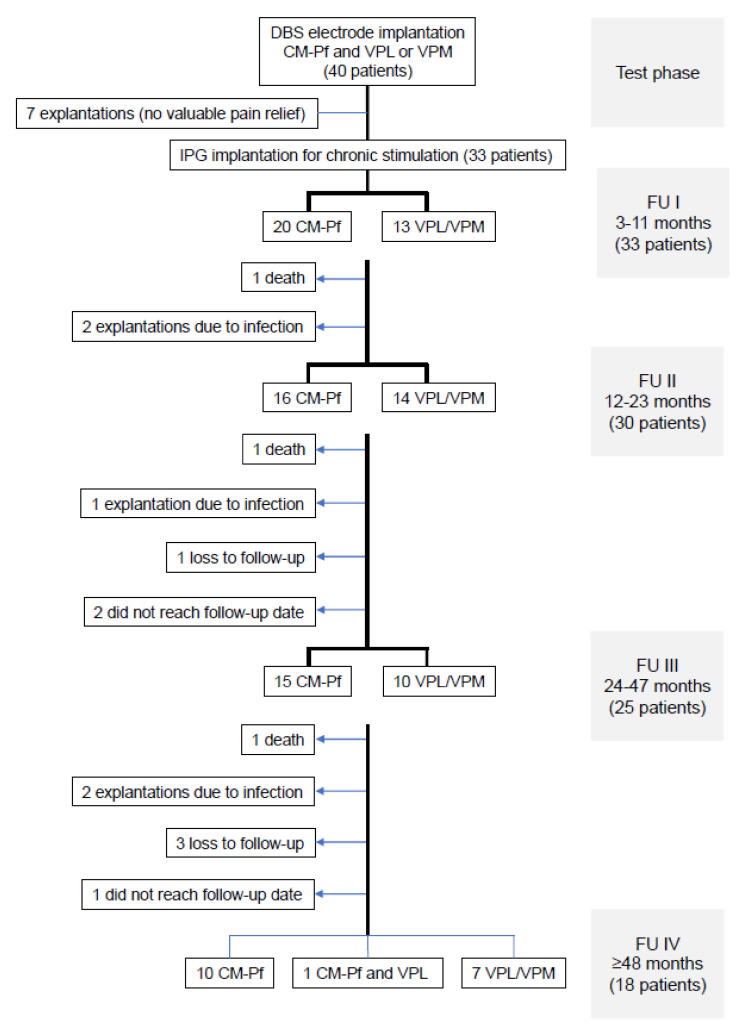
Flowchart illustrating the numbers of patients available at the different follow-up periods. The reasons for patient attrition are shown on the left side of the flow chart. CM–Pf = centromedian–parafascicular nucleus, VPL = ventroposterolateral nucleus (thalami); VPM = ventroposteromedial nucleus (thalami).

**Figure 2 biomedicines-09-00731-f002:**
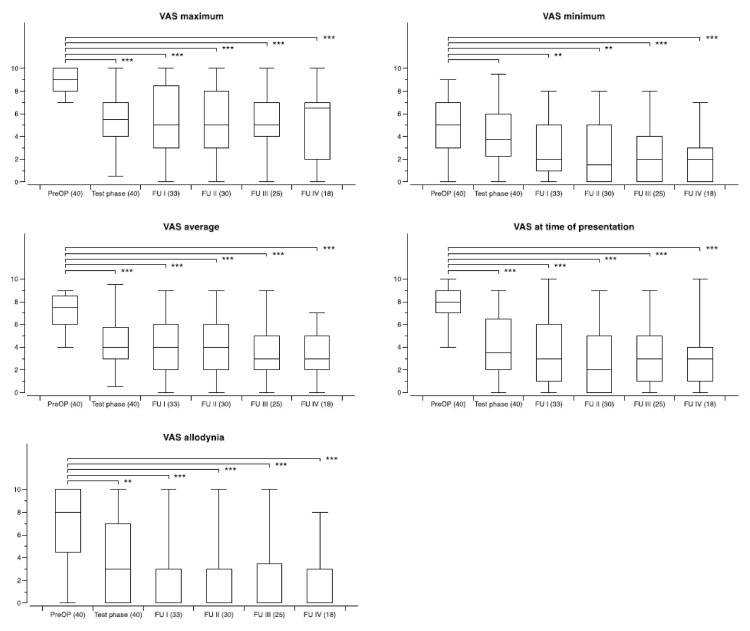
Development of specific VAS pain scores during chronic thalamic stimulation in 33 patients with various neuropathic pain syndromes. The number of patients available at the different time points is shown in brackets. Follow-up examinations were obtained at FU I: 3–12 months postoperatively, FU II: 12–23 months, FUP III: 24–47 months, and FU IV: 48-and longer. Box and whisker plots of test results over time. Boxes represent the first, second (mean), and third quartile of scores. Whiskers outline the maximum and minimum. The level of significance for the different follow-up periods as compared to preoperatively is indicated by * (* *p* < 0.05; ** *p* < 0.01; *** *p* < 0.001).

**Figure 3 biomedicines-09-00731-f003:**
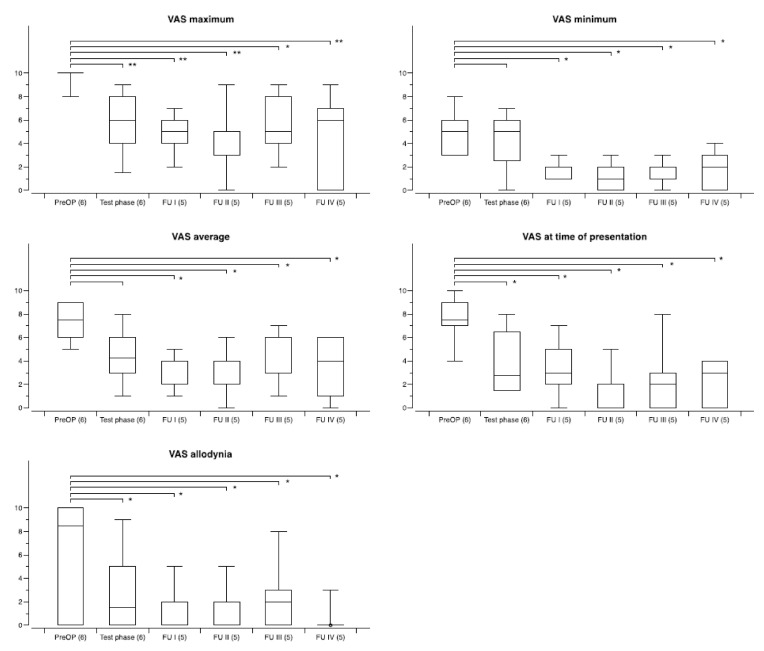
Development of specific VAS pain scores during chronic thalamic stimulation in a group of six patients with facial pain syndromes. The number of patients available at the different time points is shown in brackets. Follow-up examinations were obtained at FU I: 3–12 months postoperatively, FU II: 12–23 months, FUP III: 24–47 months, and FU IV: 48-and longer. Box and whisker plots of test results over time. Boxes represent the first, second (mean), and third quartile of scores. Whiskers outline the maximum and minimum. The level of significance for the different follow-up periods as compared to preoperatively is indicated by * (* *p* < 0.05; ** *p* < 0.01; *** *p* < 0.001).

**Figure 4 biomedicines-09-00731-f004:**
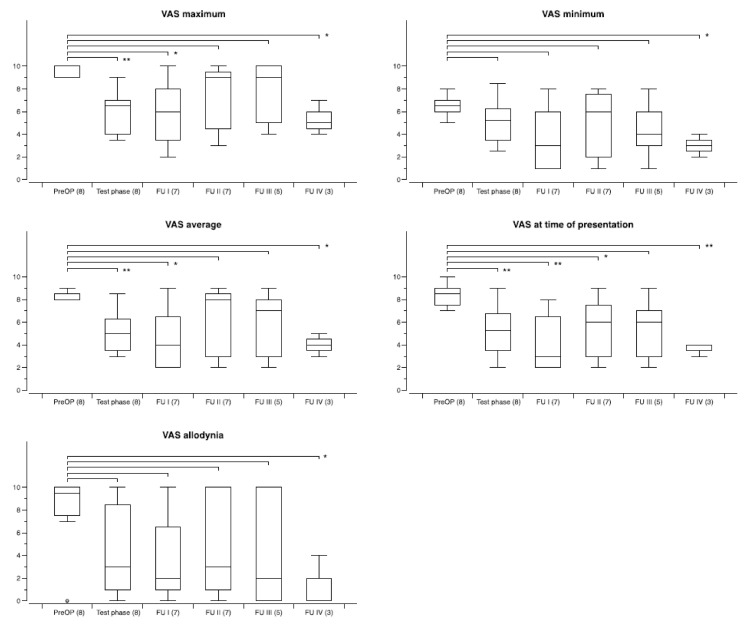
Development of specific VAS pain scores during chronic thalamic stimulation in a group of eight patients with CRPS. The number of patients available at the different time points is shown in brackets. Follow-up examinations were obtained at FU I: 3–12 months postoperatively, FU II: 12–23 months, FUP III: 24–47 months, and FU IV: 48-and longer. Box and whisker plots of test results over time. Boxes represent the first, second (mean), and third quartile of scores. Whiskers outline the maximum and minimum. The level of significance for the different follow-up periods as compared to preoperatively is indicated by * (* *p* < 0.05; ** *p* < 0.01; *** *p* < 0.001).

**Figure 5 biomedicines-09-00731-f005:**
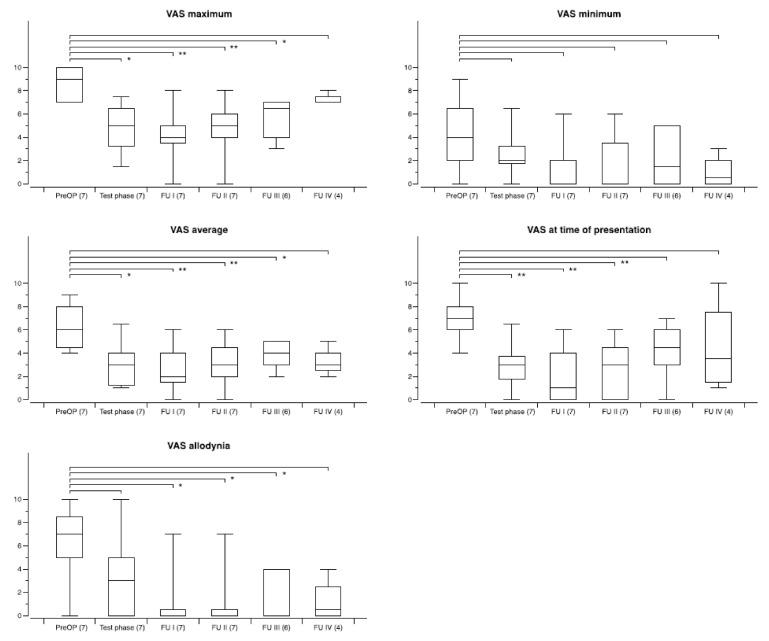
Development of specific VAS pain scores during thalamic stimulation in a group of seven patients with poststroke/central pain (except thalamus). The number of patients available at the different time points is shown in brackets. Follow-up examinations were obtained at FU I: 3–12 months postoperatively, FU II: 12–23 months, FUP III: 24–47 months, and FU IV: 48-and longer. Box and whisker plots of test results over time. Boxes represent the first, second (mean), and third quartile of scores. Whiskers outline the maximum and minimum. The level of significance for the different follow-up periods as compared to preoperatively is indicated by * (* *p* < 0.05; ** *p* < 0.01; *** *p* < 0.001).

**Figure 6 biomedicines-09-00731-f006:**
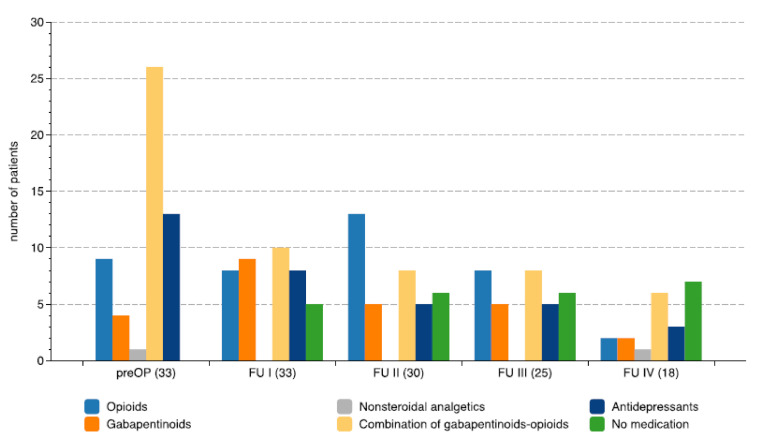
Changes in medication during chronic thalamic deep brain stimulation. The diagram shows the number of patients taking different medications at the defined follow-up periods. The number of patients available at these follow-up periods are in brackets.

**Table 1 biomedicines-09-00731-t001:** Demographic, clinical and follow-up data of 40 patients with various neuropathic pain syndromes.

	Pre-Operatively									FU I			FU II			FU III			FU IV		
Nr	Sex	Age	Diagnosis	Etiology/Category of Pain	D/Pain	Body Part Affected	VAS Max	VAS Ave	IPG	Stim Site	VAS Max	VAS Ave	Stim Site	VAS Max	VAS Ave	Stim Site	VAS Max	VAS Ave	Stim Site	VAS Max	VAS Ave
1	M	45	Poststroke/central	Brainstem infarction	2y	Face/Head	10	7	yes	CM-Pf	0	0	CM-Pf	0	0	CM-Pf	7	3	CM-Pf	7	3
2	M	72	Postherpetic	Herpes zoster	2y	Head C2 L	8	4	yes	CM-Pf	0	0	CM-Pf	0	0	CM-Pf	1	0	-	-	-
3	M	87	Postherpetic	Herpes zoster	2y	Thorax T3-6 R	10	6	yes	CM-Pf	3	0	-	-	-	-	-	-	-	-	-
4	F	50	Poststroke/ central	Multiple sclerosis	2y	Arm L	10	5	yes	CM-Pf	6	3	CM-Pf	6	3	CM-Pf	6	3	CM-Pf	8	3
5	F	67	Central thalamic	Thalamic hemorrhage	2y	Hemibody	10	9	no	-	-	-	-	-		-	-		-	-	-
6	M	52	Central thalamic	Thalamic hemorrhage	1y	Hemibody	8	6	yes	CM-Pf	5	4	-	-		-	-		-	-	-
7	F	48	Facial pain	Trigeminal nerve surgery	23y	Face R	10	7	yes	CM-Pf	2	1	CM-Pf	3	2	CM-Pf	4	3	CM-Pf	6	4
8	F	40	CRPS	Knee surgery	10y	Leg L	10	8	yes	CM-Pf	7	4	CM-Pf	5	4	CM-Pf	5	3	CM-Pf	4	3
9	M	73	Postherpetic	Herpes zoster	2y	Face R	7	6	yes	CM-Pf	0	0	CM-Pf	0	0	CM-Pf	0	0	CM-Pf	0	0
10	M	72	Phantom limb pain	Amputation	35y	Leg L	8	5	yes	CM-Pf	10	8	CM-Pf	6	5	CM-Pf	5	4	-	-	-
11	F	42	Brachial plexus injury	Motorbike accident	6y	Arm R	9	9	yes	CM-Pf	2	2	CM-Pf	3	2	CM-Pf	2	1	CM-Pf	2	1
12	F	56	Facial pain	Dental surgery	14y	Face R	10	5	yes	CM-Pf	7	4	CM-Pf	0	0	CM-Pf	2	1	CM-Pf	0	1
13	F	50	Poststroke/central	Sclerodermia	18y	Arm R	7	4	yes	CM-Pf	3	1	CM-Pf	4	2	CM-Pf	3	2	-	-	-
14	M	54	Central thalamic	Mesencephalic hemorrhage	3y	Face/Arm R	8	6	no	-	-	-	-	-		-	-		-	-	-
15	M	65	Poststroke/central	Basal ganglia hemorrhage	2y	Arm L	9	9	yes	CM-Pf	8	6	CM-Pf	8	6	CM-Pf	7	5	CM-Pf	7	5
16	M	70	Central thalamic	Thalamic infarction	10y	Hemibody	8	7	no	-	-	-	-	-		-	-		-	-	-
17	M	63	Poststroke/central	Pontomesencephalic infarction	10y	Leg R	10	9	yes	VPL	5	4	VPL	6	5	VPL	7	5	-	-	-
18	M	34	CRPS	Riding accident	5y	Hand R	10	9	no	-	-		-	-		-	-		-	-	-
19	M	33	Brachial plexus injury	Motorbike accident	14y	Arm L	9	7	yes	VPL	3	2	VPL	3	2	VPL	7	4	VPL	2	2
20	M	52	Spinal cord lesion	Car accident	4y	Hand R	8	7	no	-	-	-	-	-		-	-	-	-	-	-
21	M	65	Poststroke/ central	Frontoparietal infarction	3y	Hemibody	7	6	yes	CM-Pf	4	2	CM-Pf	4	2	-	-		-	-	-
22	F	72	Facial pain	Trigeminal nerve surgery	18y	Face R	10	9	yes	VPM	4	2	VPM	5	2	VPM	5	3	VPM	0	0
23	M	40	Spinal cord lesion	Accident	4y	Arm R	10	8	yes	CM-Pf	3	2	CM-Pf	2	1	CM-Pf	2	1	CM-Pf	10	7
24	F	58	CRPS	Ulnar nerve surgery	2y	Arm R	10	8	yes	VPL	4	2	VPL	4	2	VPL	4	2	VPL	5	4
25	M	47	Facial pain	Maxillary sinus surgery	8y	Face L	10	9	yes	VPL	5	4	VPL	5	4	VPL	8	7	VPL	7	6
26	F	24	CRPS	Injury of fingers	4y	Hand R	10	9	yes	CM-Pf	10	9	VPL	10	9	CM-Pf	10	9	-	-	-
27	F	57	CRPS	Shoulder luxation	2y	Arm R	10	8	yes	VPL	3	2	VPL	3	2	-	-		-	-	-
28	F	50	Peripheral nerve injury	Inguinal nerve surgery	3y	Thigh L	9	6	yes	VPL	7	5	VPL	5	4	VPL	8	6	Both	7	4
29	F	31	CRPS	Hand surgery	6y	Hand R	9	8	yes	CM-Pf	9	8	CM-Pf	9	8	-	-		-	-	-
30	F	44	CRPS	Ulnar nerve surgery	10y	Arm L	9	8	yes	VPL	2	2	VPL	10	9	CM-Pf	10	7	-	-	-
31	F	43	Facial pain	Dental surgery	2y	Face L	10	6	yes	VPM	6	5	VPL	9	6	VPL	9	6	VPM	9	6
32	F	68	Facial pain	Trigeminal nerve surgery	3y	Face L	8	8	no	-	-		-	-		-	-		-	-	-
33	F	26	CRPS	Accident	55y	Arm L	9	8	yes	VPL	6	5	VPL	9	8	VPL	9	8	VPL	7	5
34	M	53	Brachial plexus injury	Motorbike accident	4y	Arm R	9	6	yes	CM-Pf	10	5	CM-Pf	10	5	CM-Pf	4	3	CM-Pf	4	3
35	M	55	Poststroke/ central	Basilary artery thrombosis	5y	Hemibody	7	4	yes	VPL	5	4	VPL	5	4	VPL	4	5	VPL	7	2
36	M	49	FBSS	Spinal surgery	5y	Leg L	9	8	yes	VPL	8	6	VPL	8	5	VPL	4	2	-	-	-
37	F	70	Spinal cord lesion	Dural AV fistula embolisation	3y	Arm L	9	9	yes	VPL	10	8	VPL	7	7	-	-	-	-	-	-
38	F	69	Spinal cord lesion	Car accident	10y	Legs L/R	7	4	yes	CM-Pf	10	8	-	-	-	-	-	-	-	-	-
39	M	56	Postherpetic	Herpes zoster	3y	ThoraxT 10-11	10	9	no	-	-		-	-	-	-	-	-	-	-	-
40	F	57	FBSS	Spinal surgery	9y	Leg L	10	8	yes	CM-Pf	10	9	CM-Pf	8	7	-	-	-	-	-	-

CM–Pf = centromedian–parafascicular nucleus, VPL = ventroposterolateral nucleus (thalami); VPM = ventroposteromedial nucleus (thalami), D/pain = duration of pain, stim site = stimulation site, FBSS = failed back surgery syndrome, CRPS = complex regional pain syndrome. Note that patients with thalamic lesions are not included in the “poststroke/central” group, but listed separately as “central/thalamic”.

**Table 2 biomedicines-09-00731-t002:** Development of specific VAS pain scores during follow-up periods.

	PreOP		FU I		FU II		FU III		FU IV					
VAS	mean ± SD	n	mean ± SD	n	mean ± SD	n	mean ± SD	n	mean ± SD	n	∆_pre,I_	∆_pre,II_	∆_pre,III_	∆_pre,IV_
maximum	9.0 ± 1.1	40	5.4 ± 3.2	33	5.2 ± 3.1	30	5.3 ± 2.8	25	5.1 ± 3.2	18	−40.2%	−42.0%	−41.1%	−43.4%
minimum	4.8 ± 2.6	40	2.5 ± 2.6	33	2.5 ± 2.7	30	2.2 ± 2.2	25	2.1 ± 1.9	18	−47.4%	−48.1%	−52.8%	−55.6%
average	7.1 ± 1.6	40	3.8 ± 2.7	33	3.9 ± 2.7	30	3.7 ± 2.5	25	3.3 ± 2.1	18	−46.5%	−45.5%	−47.6%	−53.8%
presentation	7.5 ± 1.7	40	3.5 ± 3.0	33	3.1 ± 2.8	30	3.3 ± 2.7	25	3.0 ± 2.4	18	−53.2%	−66.0%	−56.1%	−59.9%
allodynia	6.9 ± 3.9	40	2.3 ± 3.5	33	2.3 ± 3.5	30	2.2 ± 3.2	25	1.8 ± 2.6	18	−66.4%	−66.9%	−67.3%	−74.0%

On the left side of the table the mean specific VAS pain values are shown for the patients available at the corresponding follow-up period. On the right side, the mean percentage of improvement is shown as compared to preoperatively, respectively. All changes were highly significant (*p* < 0.001).

**Table 3 biomedicines-09-00731-t003:** Number of responders to thalamic DBS during follow-up (VAS average pain scores).

	FU I		FU II		FU III		FU IV	
	≥50%	≥30%	≥50%	≥30%	≥50%	≥30%	≥50%	≥30%
All available patients	17/33	21/33	15/30	20/30	13/25	17/25	10/18	16/18
Facial pain	3/5	3/5	4/5	4/5	3/5	3/5	2/5	4/5
CRPS	4/7	5/7	3/7	3/7	2/5	2/5	2/3	3/3
Poststroke/central(except thalamus)	4/7	6/7	3/7	6/7	2/6	5/6	2/4	4/4
Spinal cord lesion	1/3	1/3	1/2	1/2	1/1	1/1	0/1	0/1
Brachial plexus injury	2/3	2/3	2/3	2/3	2/3	3/3	3/3	3/3
Postherpetic pain	3/3	3/3	2/2	2/2	2/2	2/2	1/1	1/1
Central thalamic	0/1	1/1	0/0	0/0	0/0	0/0	0/0	0/0
Other	0/4	0/4	0/4	2/4	1/3	1/3	0/1	1/1
FBSS	0/2	0/2	0/2	1/2	1/1	1/1	0/0	0/0
Peripheral nerve	0/1	0/1	0/1	1/1	0/1	0/1	0/1	1/1
Phantom limb	0/1	0/1	0/1	0/1	0/1	0/1	0/0	0/0

The number of responders defined by improvement of the VAS average pain score by ≥50% and by ≥30% is shown in relation to the total number of patients for whom follow-up was available in the corresponding follow-up period.

**Table 4 biomedicines-09-00731-t004:** Patient self-rating of overall benefit of chronic thalamic stimulation at the defined follow-up periods.

Self-Rating of Overall Benefit	FU I	FU II	FU III	FU IV
Excellent	6	5	6	3
Marked	12	10	7	8
Moderate	6	10	9	5
Minor	6	4	3	2
None	3	1	0	0
Total number of patients	33	30	25	18

## Data Availability

The datasets generated and analyzed during the current study are available from the corresponding author on reasonable request.

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
