# Peer review of "Centromedian–Parafascicular and Somatosensory Thalamic Deep Brain Stimulation for Treatment of Chronic Neuropathic Pain: A Contemporary Series of 40 Patients"

_biomedicines, 2021, doi:10.3390/biomedicines9070731_

Round 1

Reviewer 1 Report

It is true that nowadays the treatment of neuropathic and central pain still remains a major challenge. The treatment of chronic neuropathic pain includes physical and psychological therapies, as well as pharmacological, interventional, and surgical treatments. Opioids are often prescribed to patients who fail to achieve satisfactory pain relief with conventional treatment. Their chronic use gives tolerance as well as dependence. So, several additional pharmacological and non-pharmacological treatments have been developed and approved for chronic pain management. Thalamic deep brain stimulation (DBS) involving various target structures is a therapeutic option.

The current study was carried out on 40 patients with refractory neuropathic and central pain syndromes. They underwent stereotactic bifocal implantation of DBS electrodes in the centromedian-parafascicular and the ventroposterolateral or ventroposteromedial nucleus contralateral to the side of pain. Electrodes were externalized for test stimulation for several days.The results were assessed with five specific VAS pain scores. The authors presented a study with retrospective analysis of long term follow-up (4 years). The materials, methods and results are well presented. At discussions the authors presented the data existing in the literature concerning the subject. Through this study the authors present their experience and findings over a period of 20 years

               The authors concluded that the best results were achieved in facial pain, poststroke/ central pain (except thalamic pain), or brachial plexus injury, while patients with thalamic lesions had the least benefit.

Some remarks:

  • Lines 36-37: it is not clear what you mean by: “The mean follow-up was 62.8 months, and follow-up beyond four years”

I found the explanations in the manuscript:

  • lines 200-202: you wrote: “In order to assess long term efficacy follow-up periods were summarized as follows: FU I: early postoperative, 3-11 months; FU II: 12-23 months, FU III: 24-47 months, and FU IV: 48 months and longer. This means in your abstract the follow up beyond four years.
  • lines 260-61: you wrote: “The mean follow-up of the 33 patients who underwent subsequent IPG implantation was 62.8 months (range, 3 – 180 months)”. This means in your abstract the mean follow-up was 62.8 months.

Please clarify these 2 aspects in the abstract.

-          Lines 37-38 and lines 271-78: you wrote: “Hardware related complications requiring secondary surgeries occurred in 11/33 patients”. Unfortunately, in my opinion  the number of hardware related complications and electrode fracture is quite large. This requires hardware / electrode improvement and implicitly further studies. -          Lines 196-97: you wrote: “Assessment of outcome included rating of the five specific VAS pain scores, the MMSE and the HDRS”. At results you presented very few aspects concerning MMSE and HDRS (at lines 231-32; lines 317-320) and in Fig 6 about therapy with antidepressant. At discussion you wrote only at lines 414-15: “In addition, psychological issues and changes in mood need to be considered”.

According to Fig 6 it appears that a large number of patients take antidepressants. Therefore it would be important to follow this aspect in more detail. Is well known the fact that depression seriously affects the quality of life.

  • Lines 537-38: you wrote: “we did not include a sham stimulation protocol to control for placebo effects”. In my opinion this aspect raises a big question mark.

 In conclusion,  thalamic DBS can be a useful treatment option in selected patients with severe and medically refractory pain. It can provide also the well being of patients, facilitates their daily activities, their work productivity and improves their social relationships. These aspects prove the importance of the present study. Although the present study has some unresolved issues which requires new studies, it presents the interesting and hard work done by the authors for a period of 20 years. Also, the number of recent original publications concerning this subject is very limited and the opinions of specialists are divided. Therefore, I consider that the study is welcome and justifies the necessity to be published. For these reasons I recommend publishing this article in Biomedicine after Minor revisions.

Author Response

It is true that nowadays the treatment of neuropathic and central pain still remains a major challenge. The treatment of chronic neuropathic pain includes physical and psychological therapies, as well as pharmacological, interventional, and surgical treatments. Opioids are often prescribed to patients who fail to achieve satisfactory pain relief with conventional treatment. Their chronic use gives tolerance as well as dependence. So, several additional pharmacological and non-pharmacological treatments have been developed and approved for chronic pain management. Thalamic deep brain stimulation (DBS) involving various target structures is a therapeutic option.

The current study was carried out on 40 patients with refractory neuropathic and central pain syndromes. They underwent stereotactic bifocal implantation of DBS electrodes in the centromedian-parafascicular and the ventroposterolateral or ventroposteromedial nucleus contralateral to the side of pain. Electrodes were externalized for test stimulation for several days. The results were assessed with five specific VAS pain scores. The authors presented a study with retrospective analysis of long term follow-up (4 years). The materials, methods and results are well presented. At discussions the authors presented the data existing in the literature concerning the subject. Through this study the authors present their experience and findings over a period of 20 years

The authors concluded that the best results were achieved in facial pain, poststroke/ central pain (except thalamic pain), or brachial plexus injury, while patients with thalamic lesions had the least benefit.

Thank you for these encouraging comments.

Some remarks:

Lines 36-37: it is not clear what you mean by: “The mean follow-up was 62.8 months, and follow-up beyond four years”

I found the explanations in the manuscript:

lines 200-202: you wrote: “In order to assess long term efficacy follow-up periods were summarized as follows: FU I: early postoperative, 3-11 months; FU II: 12-23 months, FU III: 24-47 months, and FU IV: 48 months and longer. This means in your abstract the follow up beyond four years.

lines 260-61: you wrote: “The mean follow-up of the 33 patients who underwent subsequent IPG implantation was 62.8 months (range, 3 – 180 months)”. This means in your abstract the mean follow-up was 62.8 months.

Please clarify these 2 aspects in the abstract.

Thank you for pointing out this inaccuracy. The corresponding sentence in the Abstract was altered as follows:

Pacemakers were implanted in 33/40 patients for chronic stimulation for whom a mean follow-up of 62.8 months (range 3-180 months) was available. Of these, 18 patients had a follow-up beyond four years.

Lines 37-38 and lines 271-78: you wrote: “Hardware related complications requiring secondary surgeries occurred in 11/33 patients”. Unfortunately, in my opinion the number of hardware related complications and electrode fracture is quite large. This requires hardware / electrode improvement and implicitly further studies.

We agree with the reviewer´s comment. The following sentence was added to the Discussion:

While surgery was tolerated well, the long term course was burdened by hardware issues in about a third of patients which outlines the necessity of hardware and electrode improvement in future studies.

Lines 196-97: you wrote: “Assessment of outcome included rating of the five specific VAS pain scores, the MMSE and the HDRS”. At results you presented very few aspects concerning MMSE and HDRS (at lines 231-32; lines 317-320) and in Fig 6 about therapy with antidepressant. At discussion you wrote only at lines 414-15: “In addition, psychological issues and changes in mood need to be considered”.

According to Fig 6 it appears that a large number of patients take antidepressants. Therefore it would be important to follow this aspect in more detail. Is well known the fact that depression seriously affects the quality of life.

Thank you for these remarks. We completely agree. We modified the corresponding sentence as follows:

In addition, psychological issues and changes in mood need to be considered [53,54] since depression may affect markedly the quality of life independent  from pain. Notably, also at long term follow-up several patients in our study still were an antidepressant medication. This aspect, in particular needs further consideration in future studies.

Lines 537-38: you wrote: “we did not include a sham stimulation protocol to control for placebo effects”. In my opinion this aspect raises a big question mark.

We fully agree with this comment. As stated in the discussion of our manuscript this is a major limitation. Unfortunately, such a study has not become available yet.

In conclusion, thalamic DBS can be a useful treatment option in selected patients with severe and medically refractory pain. It can provide also the well-being of patients, facilitates their daily activities, their work productivity and improves their social relationships. These aspects prove the importance of the present study. Although the present study has some unresolved issues which requires new studies, it presents the interesting and hard work done by the authors for a period of 20 years. Also, the number of recent original publications concerning this subject is very limited and the opinions of specialists are divided. Therefore, I consider that the study is welcome and justifies the necessity to be published. For these reasons I recommend publishing this article in Biomedicine after Minor revisions.

Thank you for these comments.

Reviewer 2 Report

This is a fantastic series of patients studying DBS for chronic pain problems.  The one major concern I have is the high rate of infections (15%) which is concerning.  I wonder if you feel that the trial of externalization increased the risk of infections since this is higher than seen in movement disorder DBS.  Was salvage surgery ever attempted for these infection problems? 

Author Response

This is a fantastic series of patients studying DBS for chronic pain problems. 

Thank you very much for this nice comment.

The one major concern I have is the high rate of infections (15%) which is concerning.  I wonder if you feel that the trial of externalization increased the risk of infections since this is higher than seen in movement disorder DBS.  Was salvage surgery ever attempted for these infection problems? 

It is true that there was a relatively high rate of infection in our study, in particular as compared to DBS for movement disorders. The subject of externalization as a possible risk factor for infection and the prospects of salvage surgery are topics of an ongoing study at our department which, however, has not been finished yet. We added the following to the Discussion:

Furthermore, there was a relatively high rate of infection and it remains unclear whether or not temporary externalization of DBS electrodes facilitated hardware infection.